# Host Immune Defense upon Fungal Infections with Mucorales: Pathogen-Immune Cell Interactions as Drivers of Inflammatory Responses

**DOI:** 10.3390/jof6030173

**Published:** 2020-09-17

**Authors:** Dolly E. Montaño, Kerstin Voigt

**Affiliations:** 1Jena Microbial Resource Collection, Leibniz Institute for Natural Product Research and Infection Biology—Hans Knöll Institute (HKI), 07745 Jena, Germany; dolly.montano@leibniz-hki.de; 2Institute of Microbiology, Friedrich Schiller University Jena, 07743 Jena, Germany

**Keywords:** zygomycosis, phagocytosis, phagocytes, macrophages, monocytes, epi- and endothelium, Mucoromycotina, *Rhizopus*, *Rhizomucor*, *Mucor*, *Lichtheimia*, innate immune system

## Abstract

During the last few decades, mucormycosis has emerged as one of the most common fungal infections, following candidiasis and aspergillosis. The fungal order responsible for causing mucormycosis is the Mucorales. The main hallmarks of this infection include the invasion of blood vessels, infarction, thrombosis, and tissue necrosis, which are exhibited at the latest stages of the infection. Therefore, the diagnosis is often delayed, and the rapid progression of the infection severely endangers the life of people suffering from diabetes mellitus, hematological malignancies, or organ transplantation. Given the fact that mortality rates for mucormycosis range from 40 to 80%, early diagnosis and novel therapeutic strategies are urgently needed to battle the infection. However, compared to other fungal infections, little is known about the host immune response against Mucorales and the influence of inflammatory processes on the resolution of the infection. Hence, in this review, we summarized our current understanding of the interplay among pro-inflammatory cytokines, chemokines, and the host-immune cells in response to mucoralean fungi, as well as their potential use for immunotherapies.

## 1. Introduction

The life-threatening infection mucormycosis is currently the third most common invasive fungal infection [1,2]. Most of the incidences occur in patients with diabetes mellitus, hematological malignancies, organ transplantation, and in few cases also in immunocompetent individuals [1,2,3,4]. Mucormycosis is caused by fungi belonging to the order Mucorales, mainly by *Rhizopus*, Mucor, and *Lichtheimia* (ex *Absidia*) species (Figure 1) [5,6]. This infection is characterized by a fast invasion of blood vessels that further leads to infarction, thrombosis, and tissue necrosis. As a consequence, most of the patients suffer devastating results, such as massive removal of infected tissue, loss of organ functionality, and death in the worse scenario [7,8]. Contrary to candidiasis and invasive aspergillosis, mucormycosis cases are normally diagnosed in advanced stages of the infection. Furthermore, treatments against this infection are limited to liposomal amphotericin B, isavuconazole, and posaconazole, which represents a disadvantage in the case of antifungal drug resistance [5,9].

Although mucormycosis has become an emerging fungal infection, compared to the most common fungal invasive infections, aspergillosis and candidiasis, little is known about the host immune response against Mucorales [4,10]. Hence, understanding the molecular mechanisms involved in the recognition and immune response to Mucorales could provide novel insights for antifungal therapies and diagnosis. Therefore, in this review, we will summarize our current understanding of the molecular interaction among host cells in response to mucoralean fungi. In particular, we will focus on the production of cytokines and chemokines, which enables communication among leukocytes, and plays an essential role in the regulation of different cellular processes during infection, such as phagocytes recruitment, inflammation, antigen presentation, activation, and immune cell maturation [11]. Moreover, during this review, we will emphasize three relevant aspects of the host response upon Mucoralean infections: (i) the cytokine profile produced by immune cells in response to Mucorales. (ii) The influence of cytokines and chemokines on the activation of adaptive immune cells. (iii) The potential usage of cytokines as an adjuvant for immunotherapies during invasive mucormycosis. We will not address virulence factors and specific cell-mediated response to Mucorales, since these topics have been previously summarized [12,13].

## 2. Inflammation in Antifungal Response

Inflammation has been traditionally defined according to its main features: redness, warmth, pain, and swelling. However, one of the latest definition of inflammation includes its protective role in response to invading pathogens or endogenous signals, resulting in the elimination of the initial cause of injury, clearance, and tissue repair [14]. Furthermore, the activation and resolution of the inflammatory response are strictly regulated to protect the organism from collateral damages. In this regard, specialized cell-surface receptors known as pattern-recognition receptors (PRRs) activate inflammatory cascades upon the recognition of pathogen-associated molecular patterns (PAMPs) or endogenous signals, such as damage-associated molecular patterns (DAMPs) [15]. Consequently, activation of PRRs triggers the release of proinflammatory cytokines and chemokines—which are secreted proteins that exert further pro-inflammatory cascades and recruit additional immune cells to the site of infection [11]. In the case of an antifungal inflammatory-mediated response, PRRs of the host-cells recognize conserved microbial structures known as microbial associated molecular patterns (MAMPs), which are mainly found in the fungal cell-wall and comprise β-1,3-glucan, chitin, mannans, mannoproteins, and unmethylated DNA [16]. Furthermore, fungal growth and invasion by pathogenic fungi also induce local damage, promoting the release of DAMPs and amplifying the inflammatory response [17].

## 3. Clinical Features of Inflammation in Mucormycosis

Angioinvasion, thrombosis, and tissue necrosis are the most common hallmarks of mucormycosis [7]. However, inflammatory responses accompanied by these hallmarks have not been widely characterized in clinical trials. A study by Ben-Ami et al. analyzed histopathologic features in cancer patients diagnosed with pulmonary mucormycosis (PM). The authors found that 100% of the patients with PM showed angioinvasion; on the contrary, only 30% of the patients exhibited inflammatory infiltrates, such as aggregates of macrophages in the margin of necrotic tissues [18]. Furthermore, patients within the same cohort that underwent allogeneic hematopoietic stem cell transplantation (HSCT) had more inflammatory cell infiltration compared to non-HSCT patients [18]. Similarly, most of the mucormycosis incidences in patients with diabetes as primary disease exhibited inflammatory infiltration of neutrophils, multinucleated giant cells, and phagocytes in the site of infection, as well as tissue damage and necrosis [19,20,21]. Therefore, we postulate that the recruitment of inflammatory cells during mucormycotic infections is likely dependent on pre-existing primary diseases in the patients. Nevertheless, more studies are needed to understand the correlation among these clinical observations.

## 4. Common Pro-Inflammatory Cytokines and Chemokines during Mucoralean Infection

During infection by diverse mucoralean species, leukocytes communicate with each other by producing a wide range of cytokines and chemokines (Table 1). Interleukin-1 beta (IL-1β) has been described to play an important role in response to pathogenic fungi [22,23]. This interleukin is involved in the induction of other pro-inflammatory proteins, hematopoiesis, differentiation of TH17 cells, and development of IL-10 [11]. The tumor necrosis factor α (TNF-α) mediates the host-apoptosis and inflammation, where it has a dual role as an activator and immunosuppressor of the inflammatory response [24]. Another common interleukin is IL-6, which mediates trafficking of leukocytes, induces the production of acute-phase proteins, promotes T-cell proliferation, B-cell differentiation, survival, and plasma cell production of IgG, IgA, and IgM [25]. Similarly, the interleukin IL-12 promotes the development and maintenance of Th1 cells, the activation of natural killer (NK) cells, and the maturation of dendritic cells (DCs) [11]. IL-12 also controls the production of interferon-γ (IFN-γ), which is a major product of Th1 cells and induces antiviral responses, cellular growth, apoptosis, leukocyte trafficking, and activation of macrophages [26]. Furthermore, an effective immune response requires the recruitment of leukocytes to the side of infection by chemoattractant proteins, such as interleukin-8 (IL-8), which mediates the recruitment of neutrophils, NK cells, T cells, basophils, and eosinophils [27]. Meanwhile, the monocyte chemoattractant protein-1 (MCP-1/CCL2) regulates the trafficking of monocytes/macrophages to the infected tissue [28]. Finally, the granulocyte-macrophage colony-stimulating factors (GM-CSF) enhance survival, activation, and differentiation of monocyte/macrophages, neutrophils, and eosinophils [29].

## 5. Cytokine Modulation by Polymorphonuclear Neutrophils (PMNs)

Polymorphonuclear neutrophils (PMNs) belong to the first line of defense in the innate immune response and play an essential role in controlling fungal infections. These cells rapidly migrate into the infected tissue, produce reactive oxygen species, and release neutrophil extracellular traps (NETs) [42]. Furthermore, PMNs modulate the host immune response against *R. oryzae*, *R. microsporus*, and *L. corymbifera* by producing IL-1β, TNF-α, and IL-8 [30,31]. Interestingly, PMNs treated with a combination of IFN-γ and GM-CSF increased the production of TNF-α in response to *R. microsporus* and *L. corymbifera*, while incubation with only IFN-γ suppressed PMNs’ ability to release IL-8 after infection with these fungi [30]. The treatment with IFN-γ and GM-CSF did not only modulate the cytokine production of PMNs but did also increase the percentages of hyphal damage to *R. oryzae*, *R. microsporus*, and *L. corymbifera* [30]. The immune response mediated by PMNs differs among different fungal species [31]. In this regard, a pioneering study revealed that PMN-mediated fungal activity against *R. arrhizus* sporangiospores was significantly lower compared to *A. fumigatus* and *C. albicans*, whereas G-CSF administration to healthy donors enhanced by fifteen-fold PMNs-mediated killing of *R. arrhizus* [43]. Moreover, Andrianaki et al. showed that PMNs from immunocompetent mice have a lower phagocytic rate of *R. oryzae* compared to *A. fumigatus* [44]. Based on these observations, we hypothesize that the effectivity of PMNs-mediated antifungal response could be influenced by the cytokines profile in response to different fungal species.

## 6. Macrophages Inflammatory Response to Mucorales

Although tissue-resident macrophages are one of the first immune cells that recognize and clear fungi, little is known about the inflammatory mechanisms driven by macrophages in response to Mucorales [45]. In 2018, Lopez et al. developed an adult zebrafish model of *Mucor circinelloides* infection. In this study, the authors revealed 857 differentially expressed genes after infection with this mucoralean fungus. Several of these genes were involved in the regulation of immune responses; for example, the mRNA levels of IL-1β, TNF-α, IL-22 were increased in response to germination and invasion by *M. circinelloides* [32]. These observations were additionally confirmed in murine J774 macrophages. Moreover, infection with *M. circinelloides* depleted neutrophils and macrophages in the kidney, the major hematopoietic organ in zebrafish [32]. In addition, a recent study revealed the role of the heat-shock protein A8 (Hspa8) in the recognition of *L. corymbifera* by murine alveolar macrophages [46]. In this study, the authors showed by LC-MS/MS, colocalization analyses, and blocking antibodies, that Hspa8 plays a role in the recognition of *L. corymbifera* and opens new venues for therapeutic approaches against this fungus [46].

Iron homeostasis has been closely related to the pathogenicity of different fungal species and the modulation of the inflammatory response by macrophages [44,47,48].In addition to eradicating pathogens, macrophages play an important role in iron-recycling from senescent erythrocytes, a process by which the iron transporter ferroportin (Fpn1) is crucial [49]. The deletion of Fpn1 in murine macrophages led to the accumulation of iron in vital organs and enhanced expression of the pro-inflammatory cytokines TNF-α and IL-6 [49]. Interestingly, pro-inflammatory cytokines also affect the iron homeostasis of macrophages by blocking the intracellular traffic of this metal, resulting in a host nutritional strategy to reduce its availability for pathogens [48]. In this regard, Adrianaki et al. observed that iron starvation inside alveolar macrophages inhibited *R. oryzae* growth despite its intracellular persistence [44]. Since iron homeostasis and the inflammatory response are mutually regulated [48], more studies in this regard could provide a better understanding of the host-pathogen nutritional interplay.

## 7. Immune Response to Mucorales by Peripheral Blood Mononuclear Cells

Peripheral blood mononuclear cells (PBMCs) are constituted by approximately 80% of T and B cells, 10 % of natural killer cells, and 10 % of monocytes, and all of these leukocytes play an essential role in both the innate and the adaptive immune response [50]. Although different leukocytes induce a pro-inflammatory profile against pathogenic fungi, the amount and type of cytokine produced by these cells may vary in response to different fungal species. For example, mucoralean fungi induce the production of the pro-inflammatory cytokines TNF-α, IL-1β, IL-6, IL-8, MCP-1, and GM-CSF by PBMCs after nine hours of co-incubation with inactivated *R. arrhizus*, *C. bertholletiae*, *M. circinelloides*, *M. hiemalis*, *L. corymbifera*, *Rh. pusillus*, and *R. microsporus* spores and germ tubes. Interestingly, the production of IL-1β in response to these Mucorales was significantly higher compared to *A. fumigatus* [33]. Furthermore, Warris et al. also described a similar pattern, where concentrations of IL-6 and TNF-α in response to *R. oryzae* were significantly higher compared to *A. fumigatus* stimulus [51]. Many studies on antifungal cytokines production were performed with PBMCs. However, when evaluating the specificity of each cell type, the complexity of these set of leukocytes do not reflect their individual role in the antifungal response. Nonetheless, the interaction among PBMCs during infection provides a closer approximation to conditions found in vivo, where leukocytes have crosstalk via cytokine production.

## 8. Immune Response to Mucorales by Monocyte-Derived Dendritic Cells and Epithelial Cells

Dendritic cells (DCs) are known as “the bridge” between the innate immune response and the adaptive immune response due to their ability to phagocyte pathogens and present their antigens to naive T cells, which results in further immunological memory to that specific pathogen [52]. Therefore, DCs are thought to play an important role in the antifungal immune response. Wuster et al. in 2017 investigated the influence of pathogenic fungi on the maturation of DCs by evaluating the expression of co-stimulatory molecules—cell surface molecules that modulate T cells’ activation [52]. The authors found that after eighteen hours of co-culture with germ tubes and resting spores of *R. arrhizus*, the co-stimulatory molecules CD83 and CD86 were significantly upregulated on monocyte-derived dendritic cells (moDCs), while incubation with *A. fumigatus* spores did not induce a strong upregulation of the co-stimulatory molecules [33]. These observations suggested that *R. arrhizus* promotes maturation of moDCs, which might translate into a robust induction of adaptive immunity.

DCs reside on tissues, and thus, their response against infection is likely influenced by other cells in the microenvironment. In this regard, Belic et al. developed an alveolar bilayer model to study the interaction of epithelial cells and moDCs during the invasion by *R. arrhizus*, *C. bertholleitiae*, and *R. pusillus* [34]. In this experimental setup, the authors cultivated human pulmonary endothelial cells-HPAEC, epithelial cells-A549, and moDCs in a trans-well system. In the lower compartment, they placed endothelial cells, and in the upper compartment, they cultivated epithelial cells, moDCs, and the mucoralean fungi for thirty hours. After infection with the pathogenic fungi, they found that the production of IL-1β, IL-6, IL-8, IL-12, TNF-α, CCL2, and CCL5 was significantly increased after the addition of moDCs. Interestingly, epithelial cells also produced IL-6 and IL-8 in absence of moDCs and independently of stimuli with the fungi. On the contrary, infected epithelial cells had reduced production of the chemokines CCL2 and CCL5 in the absence of moDCs [34]. Altogether, the authors showed the major role of moDCs on the inflammatory cytokine and chemokine secretion during an in vitro infection model with pathogenic Mucorales.

## 9. Natural Killer (NK) Cells in Response to Mucorales

NK cells represent about 10–15% of PBMCs and exhibit direct antifungal activity by releasing cytotoxic molecules such as perforin or granzyme B [53,54,55]. In addition, NK cells exert indirect antifungal activity by modulating immune cells via the production of diverse cytokines, including IFN-γ, TNF-α, GM-CSF, and CCL5, [53,54,56]. Particularly, NK cells can cause hyphal damage to *R. oryzae* by releasing perforin but do not affect the viability of fungal resting sporangiospores [35]. However, this pathogenic mucoralean species exhibits immunosuppressive effects on NK cells by reducing the production levels of IFN-γ and the pro-inflammatory chemokine RANTES (CCL5) [35], which promotes chemotaxis and migration of T cells and monocytes to the site of infection [57]. Furthermore, NK cells pre-stimulated with IL-2 induced cell-mediated damage on clinical isolates of *L. ramosa*, *L. corymbifera*, *M. circinelloides*, and *R. microsporus*, with *L. corymbifera* being among the most affected Mucorales. In addition, all mucoralean clinical isolates induced similar levels of IFN-γ secretion by NK cells [36]. Since different pathogenic mucoralean fungi induce similar and effective antifungal responses by NK cells, elucidating the receptors on NK cells that recognize these fungi would contribute to our understanding of the host-pathogen interaction and the potential development of new therapeutic strategies.

## 10. Production of Cytokines by T Cells in Response to Mucorales

Immunological memory refers to the ability of our immune system to recognize and respond to a previously encountered antigen. The adaptive immune response accomplishes this function through specific cells known as T lymphocytes [58]. These cells also contribute to the maintenance, self-tolerance, and homeostasis of the immune system [58]. T cells are classified into (1) cytotoxic T cells (CD8^+^) and (2) helper T cells (CD4^+^) that differentiate into specific subsets including T helper 1 (Th1), Th2, Th17, and regulatory T cells (Treg) [59]. Each subset of Th cells possesses a specific cytokine profile that modulates anti-fungal immunity [59,60]. In particular, Th1 releases IFN-γ, TNF-α, and GM-CSF, which foster the activation of macrophages and neutrophils. Th2 modulates B cell function via IL-4, IL-5, and IL-13, production. Treg releases the immunosuppressive cytokines IL-10 and TGF-β and Th17 produces IL-17—an important cytokine involved in the defense against fungal infections [60,61]. The development of these subsets is mainly determined by the cytokine prevailing in the microenvironment during the antigen-presentation [62]. During Th polarization, the prevailing cytokine has an inhibitory effect on the development of other types of Th phenotypes, which increases the immune response of each subset at a particular time and condition [62]. For example, Th1 development is induced by the pro-inflammatory cytokine IL-12 released by antigen-presenting cells [60]. Previous studies have shown that mutations in the IL-12 signaling pathway exert a predisposition to fungal diseases, such as cryptococcosis, candidiasis, paracoccidioidomycosis, and coccidioidomycosis [61]. Furthermore, IFN-γ produced by Th1 cells protects against histoplasmosis, aspergillosis, cryptococcosis, and coccidioidomycosis. On the other hand, the absence of this cytokine increased susceptibility to these fungal infections in mice and humans. Moreover, adjunctive immunotherapy with IFN-γ augmented protection against the fungi [61]. In addition, Th17 has been well characterized in anti-fungal immunity: particularly against *C. albicans*. Here, the cytokines (IL-1β, IL-6, IL-23) and TGF-β are required for priming Th17 differentiation, expansion, and production of IFN-γ and IL-17 [62,63].

Regarding Th responses to mucormycosis, Chamilos et al. found that *Rhizopus* hyphae trigger IL-23 production by DCs via Dectin-1 activation, which drives Th17 responses [40]. A more recent study revealed that Card9−/− mice exhibited impaired local cytokine production (IL-17, IFN-γ, and TNF-α) and Th1 cell response to *R. arrhizus* in a murine infection model [64]. Furthermore, stimulation with spores and germ tubes of *R. arrhizus* induced the upregulation of the glycoprotein CD154 (an indicator of immune cell activation) in CD4^+^ T cells of healthy donors [33]. Similarly, Mucorales-specific T cells were detected in hematological patients with invasive mucormycosis (IM), whereas patients with invasive infections other than IM did not have significant frequencies of specific-T cells [38]. Moreover, in the patients with hematological malignancies, T cells produced a characteristic cytokine profile with IL-10 secreted during the initial stages of the infection, followed by IFN-γ and IL-4 production at later time points [38]. On the contrary, another study showed that T cells from healthy donors have a low reactivity to *R. oryzae* and low IFN-γ production [37]. These discrepancies may rely on the methods used for the evaluation of specific-T cell activity. While some authors measured CD154 expression as an indicator of T cell activation, the latter assessed cytokine secretion after exposure to the fungi. Nevertheless, each approach may provide helpful information about different aspects of T cell functionality, either during interaction with antigen-presenting cells or by regulating the immune response via cytokine production.

T cells are particularly interesting due to their potential use for immunotherapy and diagnostics. Recently, Castillo et al. found that the treatment of T cells with a combination of IL-2/IL-7 induce a proper expansion of *R. oryzae*-specific T cells with a strong production of IL-13, IL-5, TNFα, and IL-10. Therefore, the authors considered the expansion of rare Mucorales-specific T cells from healthy donors to be used for potential adoptive immunotherapy [37]. Page et al. evaluated Mucorales-specific T cells and cytokine profiles in healthy donors as biomarkers of environmental mold exposure. They found higher CD154+/CD4+ T cell frequencies in a cohort of healthy donors that were highly exposed to environmental molds, such as *R. arrhizus*, *R. pusillus*, and *C. bertholletiae*, compared to donors with none or low exposure [39]. Moreover, the authors indicated a significant correlation between *A. fumigatus* and Mucorales specific-T cell numbers and suggested a possible cross-reactivity of T cells against multiple fungi. In this regard, cytokine profiles of highly exposed subjects showed more than two-fold increased production of IFNγ, TNF-α, IL-5, IL-1β, and IL-17A by PBMCs in response to *R arrhizus.* However, statistical significance was observed only for IL-5, contrary to the *A. fumigatus* stimulus which also exhibited significant elevation of IL-13, and IL-17A [39].

## 11. B Cell Lymphocytes and Humoral Response against Mucorales

In contrast to T cells, B cells undergo clonal expansion and secrete specialized proteins known as antibodies (immunoglobulins) upon activation [61]. These antibodies bind to antigens expressed on the surface of pathogenic microorganisms and exert anti-microbial activity via neutralization, opsonization, complement activation, and antibody-dependent cellular cytotoxicity [61]. The importance of immunoglobulins in antifungal immunity has been controversially discussed. While some pioneering studies in the field claimed that antibodies do not play a major role in antifungal clearance, more recent studies showed that immunoglobulins confer protection against *Cr. neoformans*, *C. albicans*, *Pneumocystis* spp., and *H. capsulatum* [65,66,67,68]. Nevertheless, studies on the B cell-mediated antifungal response are still scarce. The spore protein CotH3 binds to the GRP78 receptor in endothelial cells and was reported to contribute in the humoral response against Mucorales [69]. Polyclonal antibodies raised against peptides of CotH3 protected mice under neutropenia and diabetic ketoacidosis from *R. delemar*, *Rhizomucor*, *Apophysomyces*, *Lichtheimia*, *Cunninghamella, and M. circinelloides* [69].

## 12. The Tissue Microenvironment and Its Role in the Inflammatory Response

Microenvironmental factors other than cytokines (e.g., oxygen, lipids, glucose, microbial products, or salts) are involved in the differentiation of immune cells, which might impact their capacity to control fungal infections [70,71,72,73,74]. For example, sodium chloride (NaCl) is usually found in peripheral tissues and promotes differentiation of human and murine Th17 cells with anti-inflammatory properties [70,75]. The NaCl effect is infection-dependent as shown by the promotion of anti-inflammatory functions in *Staphylococcus aureus*-specific Th17 cells but not in *Candida*-specific Th17 cells [75]. On the contrary, this NaCl anti-inflammatory effect was reversed by the cytokine IL-1β. In the context of Mucorales stress response to the host-microenvironment, *R. arrhizus,* and other Zygomycetes (former class Phycomycetes) exhibited stress tolerance after exposure to high NaCl concentrations [76,77]. Therefore, we hypothesize that a series of factors, such as NaCl-mediated anti-inflammatory responses, high osmotic tolerance of Mucorales, and the disbalance of NaCl metabolism in people with risk to develop mucormycosis (e.g., diabetes) may create an optimal microenvironment for cutaneous mucormycosis [78,79,80]. Nevertheless, more studies are needed to understand the contribution of sodium chloride in shaping immune responses in the pathogenesis of the Mucorales.

## 13. Cytokines for Immunotherapies in the Battle against Mucormycosis

Currently, effective treatments against mucormycosis are based on four fundamental aspects: early diagnosis, formulation of liposomal amphotericin B, surgical debridement of infected tissue, and reversal of immunosuppression [9,81]. Hence, some authors suggested that certain cytokines could contribute to reverse immunosuppressive conditions in patients with mucormycosis [30,81,82]. As an example, treatment with IFN-γ and GM-CSF increased TNF-α production by PMNs in response to *R. microsporus* and *L. corymbifera* [30]. Furthermore, these cytokines increased the percentage of PMN-mediated hyphal damage to *R. oryzae*, *R. microsporus*, and *L. corymbifera* [30]. Moreover, IL-2 in combination with IL-7 induced expansion of *R. oryzae*-specific T cells in healthy donors [37]. These findings support initiatives for cytokine-mediated adoptive immunotherapy, which could either boost different cellular-types of the innate immune response or promote the expansion of Mucorales-specific T cells. However, the mechanisms by which these cytokines modulate the immune response against Mucorales remain poorly understood. Therefore, more studies and clinical reports in the context of inflammation and cytokine modulation are needed in order to weigh potential benefits against possible complications, including hyperinflammation, anaphylaxis, or prolonged anti-inflammatory responses that may potentially expose the host to secondary infections.

## 14. Future Perspectives

On balance, inflammation plays a crucial role in the resolution of fungal infections by alerting the host-immune system of external threats and promoting the clearance of the pathogenic fungi [14,15]. As mentioned throughout this review, recognition of mucoralean fungi by diverse host immune cells triggers inflammatory processes that are regulated by specific cytokines (Figure 2). Hence, we considered it beneficial to our current knowledge to perform more studies on cytokines-mediated immune response against mucoralean fungi. Although our understanding of the host-inflammatory response against Mucorales has increased during the last decades—especially in response to *Rhizopus* and *Mucor* species—the molecular mechanisms involved in the production of pro-inflammatory cytokines and the mucoralean-specific antigens triggering the inflammatory response remain unknown. Understanding this piece of the puzzle could open up new perspectives for novel therapeutic strategies and diagnosis, such as: boosting immune-cell function against mucoralean fungi, identifying specific cytokine-profiles in response to Mucorales, improving adoptive cell transfer, or suppressing hyperinflammatory processes during infection. However, all these strategies must be carefully evaluated in the context of clinical manifestations and pre-existing conditions of patients suffering from mucormycosis—including diabetes, neutropenia, solid organ transplants, and hematological malignancies—to reduce deleterious effects in these patients.

## Figures and Tables

**Figure 1 jof-06-00173-f001:**
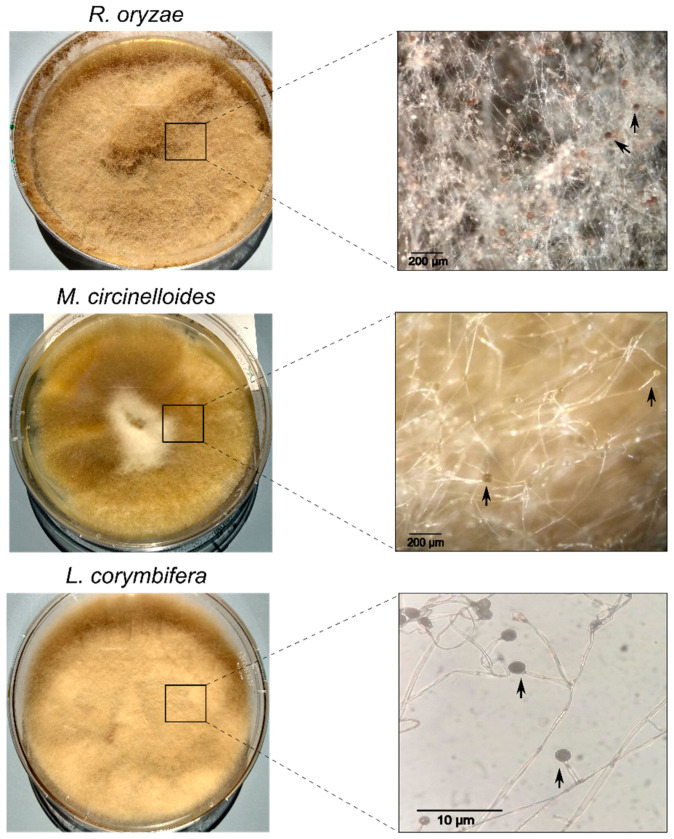
The most common pathogenic mucoralean fungi. *Rhizopus oryzae* (syn.: *Rhizopus arrhizus*), *Mucor circinelloides*, and *Lichtheimia corymbifera* are the most frequently reported causative agents of mucormycosis. The left panel depicts the colony morphologies of four-week-old *R. oryzae* and *L. corymbifera* cultivated on malt extract agar, as well as the colony morphology of one-week-old *M. circinelloides* cultivated on potato dextrose agar. The right panel represents the corresponding micromorphologies including sporangia (marked with arrows), in which the sporangiospores are produced. Sporangium of *R. oryzae* presents a dark-brown coloration, while *M. circinelloides* exhibits a yellow color and *L. corymbifera* a dark grey pigmentation.

**Figure 2 jof-06-00173-f002:**
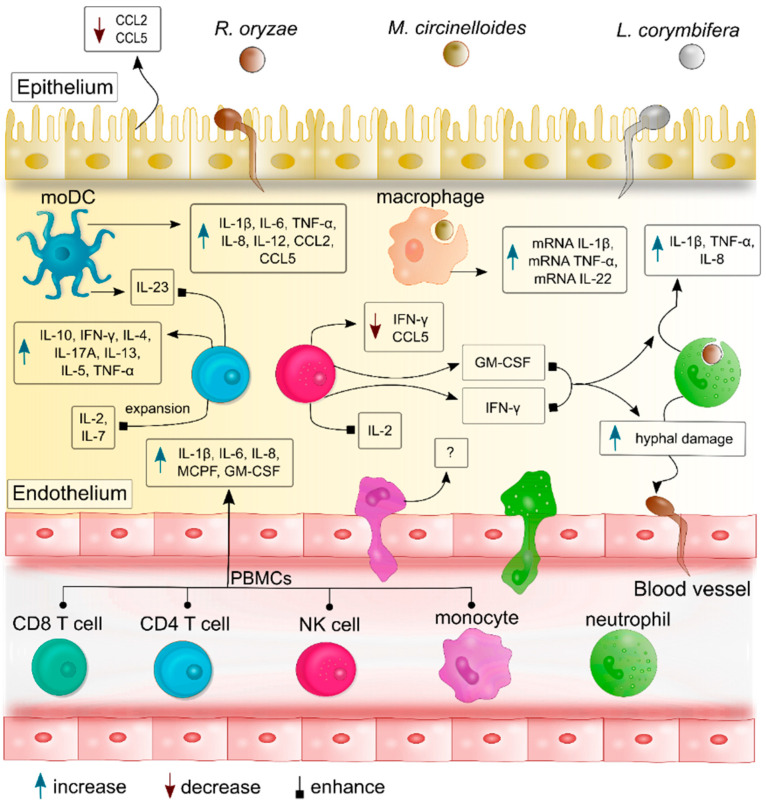
Cytokines and chemokines profiles produced by different host immune cells during mucoralean fungal infection. After recognition of germling and spores of the most common pathogen mucoralean fungi—*R. oryzae* (syn.: *R. arrhizus*), *M. circinelloides*, and *L. corymbifera*—immune cells produce specific cytokines and chemokines that promote activation and expansion of other immune cells, as well as additional recruitment of leukocytes from the bloodstream to the site of infection. These cytokines regulate the inflammatory response by exerting further pro-inflammatory cascades and promoting communication among the immune cells. As an example, GM-CSF and IFN-γ produced by NK cells and T cells enhance the cytokine production of neutrophils, and their percentages of hyphal damage to *R. oryzae*, and *L. corymbifera.* Meanwhile, IL-2 and Il-7 induce the expansion of Mucorales-specific T cells and their production of IL-13, IL-5, TNFα, and IL-10. Moreover, DCs produce IL-23 which promotes the release of IL-17A by T cells.

**Table 1 jof-06-00173-t001:** Cytokine and chemokine production in response to pathogenic mucoralean species—an overview of current studies.

Categories of the Immune Response	Cellular Type	Cytokines/Chemokines	Mucorales Species	Reference
Innate immune response	polymorphonuclear neutrophils (PMNs)	IL-1β, TNF-α, IL-8	*R. oryzae* *, *R. microsporus*, *L. corymbifera*	[30,31]
		IFN-γ and GM-CSF enhanced IL-1β, TNF-α, IL-8 production	*Rhizopus microsporus*, *L. corymbifera*	[30]
	Macrophages	mRNA of interleukin-1β (il1b), tumor necrosis factor α (tnfa) and il22	*M. circinelloides*	[32]
	Peripheral blood mononuclear cells (PBMCs)	TNF-α, IL-1β, IL-6, IL-8, MCP-1, and GM-CSF	*R. arrhizus* * *Cunninghamella bertholletiae*, *M. circinelloides*, *M. hiemalis*, *L. corymbifera*, *Rhizomucor pusillus*, *R. microsporus*	[33]
	Epithelial cells	Reduced CCL2 and CCL5	*R. arrhizus* *, *C. bertholletiae*, *Rh. pusillus*	[34]
	Dendritic cells	IL-1β, IL-6, IL-8, IL-12, TNF-α, CCL2, and CCL5	*R. arrhizus* *, *C. bertholletiae*, *Rh. pusillus*	[34]
	Natural killer (NK) cells	reduced levels of IFN-γ and RANTES (CCL5)	*R. oryzae* *	[35]
		IFN-γ	*Lichtheimia ramosa*, *L. corymbifera*, *M. circinelloides*, *R. microsporus*	[36]
Adaptive immune response	T cells	reduced IFN-γ production	*R. oryzae* *	[37]
		IL-10, IFN-γ, and IL-4 in hematological malignances	Non-specified Mucorales	[38]
		IL-2/IL-7 induce expansion of specific T cells	*R. oryzae* *	[37]
		IL-13, IL-5, TNFα, and IL-10	*R. oryzae* *	[37]
		IFN-γ, TNF-α, IL-5, IL-1β, and IL-17A	*R. arrhizus* *	[39]
		IL-17A induced by IL-23	*R. arrhizus*	[40]

IL-1β: interleukin-1 beta; TNF-α: tumor necrosis factor α; IFN-γ: interferon-γ; IL-8: interleukin-8; GM-CSF: granulocyte-macrophage colony-stimulating factors. * *R. oryzae* is considered as a synonym of *R. arrhizus* [41]. However, both names have been cited according to the original references.

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
