# Peer review of "Host Immune Defense upon Fungal Infections with Mucorales: Pathogen-Immune Cell Interactions as Drivers of Inflammatory Responses"

_jof, 2020, doi:10.3390/jof6030173_

Round 1

Reviewer 1 Report

The authors present a concise review of the current knowledge regarding immunological defense mechanisms against Mucorales.  The authors themselves state in the abstract that relatively little is known about immune responses to Mucorales compared to other fungal infections, but nevertheless this review covers a great deal of literature in an organized manner.  The English is overall quite good, but there are a few typos and awkward constructions that a good proof reading will correct.  While Figure 1 does not really add much to the review, Figure 2 is very nice and highlights the complexity of what is known in the field.  

The one weakness of the review, in this Reviewer's opinion, is the discussion of T cell responses.  This deserves to be highlighted a bit more by introducing what is known about protective T cell responses to fungal pathogens in general (ie in terms of Th subset polarization and cytokines) and how this pattern may or may not relate to Mucorales in particular, based on the findings that are discussed.  This discussion would also help readers better appreciate the basis of any 'cytokine-mediated adoptive immunotherapy' mentioned in lines 282-284.  A more comprehensive discussion of T cell responses in the context of anti-fungal immunity in general would thus strengthen the review.

Finally, it may be worth at least mentioning B cells and antibodies and any putative role(s) for these during fungal infection in general as this arm of the immune response is not discussed, and this could be viewed as an oversight by the authors by a broader readership.

Author Response

1. There is a discrepancy between the title and the actual content The English is overall quite good, but there are a few typos and awkward constructions that a good proof reading will correct.

We have eliminated the following sentence constructions and typos (page and line nos. refer to the manuscript including track changes):

  • Page 1, lines 20, 29, 31.
  • Page 2, lines 45-46, 52.
  • Page 3, lines 72-77, 80, 87, 89-90.
  • Page 4, lines 108, 113, 115-116.
  • Page 5, Table 1, T cells section.
  • Page 6, lines 133-135, 149, 153-154, 157-163.
  • Page 7, lines 174-177, 183, 202.
  • Page 8, lines 217-219, 230-232, 240.
  • Page 9, lines 245-266, 276-272, 284, 285.
  • Page 10, line 291, 292, 301-113, 316-325.
  • Page 11, lines 331, 335, 336, 343, 344.
  • Page 12, lines 378, 379.

All changes have been highlighted with the Track Changes function of Microsoft Word.

2. The one weakness of the review, in this Reviewer's opinion, is the discussion of T cell responses. This deserves to be highlighted a bit more by introducing what is known about protective T cell responses to fungal pathogens in general (ie in terms of Th subset polarization and cytokines) and how this pattern may or may not relate to Mucorales in particular, based on the findings that are discussed.  This discussion would also help readers better appreciate the basis of any 'cytokine-mediated adoptive immunotherapy' mentioned in lines 282-284.  A more comprehensive discussion of T cell responses in the context of anti-fungal immunity in general would thus strengthen the review.

We are thankful to the reviewer for this comment. We have included one and a half paragraph in the 10th chapter (entitled ‘Production of cytokines by T cells in response to Mucorales’) addressing the role of cytokines in Th subset polarization, the role of T cell immunity against fungal pathogens, and the current knowledge on Th responses to mucormycosis. Pages 8-10, from lines 246-275:

Immunological memory refers to the ability of our immune system to recognize and respond to a previously encountered antigen. The adaptive immune response accomplishes this function through specific cells known as T lymphocytes [58]. These cells also contribute to the maintenance, self-tolerance, and homeostasis of the immune system [58]. T cells are classified into (1) cytotoxic T cells (CD8+) and (2) helper T cells (CD4+) that differentiate into specific subsets including T helper 1 (Th1), Th2, Th17, and regulatory T cells (Treg) [59]. Each subset of Th cells possesses a specific cytokine profile that modulates anti-fungal immunity [59, 60]. In particular, Th1 releases IFN-γ, TNF-α, and GM-CSF which foster the activation of macrophages and neutrophils. Th2 modulates B cell function via IL-4, IL-5, and IL-13, production. Treg releases the immunosuppressive cytokines IL-10 and TGF-β and Th17 produces IL-17—an important cytokine involved in the defense against fungal infections [60, 61]. The development of these subsets is mainly determined by the cytokine prevailing in the microenvironment during the antigen-presentation [62]. During Th polarization, the prevailing cytokine has an inhibitory effect on the development of other types of Th phenotypes, which increases the immune response of each subset at a particular time and condition [62]. For example, Th1 development is induced by the pro-inflammatory cytokine IL-12 released by antigen-presenting cells [60]. Previous studies have shown that mutations in the IL-12 signaling pathway exert predisposition to fungal diseases, such as cryptococcosis, candidiasis, paracoccidioidomycosis, and coccidioidomycosis [61]. Furthermore, IFN-γ produced by Th1 cells protects against histoplasmosis, aspergillosis, cryptococcosis, and coccidioidomycosis. On the other hand, the absence of this cytokine increased susceptibility to these fungal infections in mice and humans. Moreover, adjunctive immunotherapy with IFN-γ augmented protection against the fungi [61]. In addition, Th17 has been well characterized in anti-fungal immunity: particularly against C. albicans. Here the cytokines (IL-1β, IL-6, IL-23) and TGF-β are required for priming Th17 differentiation, expansion, and production of IFN-γ and IL-17 [62, 63].

Regarding Th responses to mucormycosis, Chamilos et al. found that Rhizopus hyphae trigger IL-23 production by DCs via Dectin-1 activation, which drives Th17 responses [40]. A more recent study revealed that Card9−/− mice exhibited impaired local cytokine production (IL-17, IFN-γ, and TNF-α) and Th1 cell response to R. arrhizus in a murine infection model [64]. Furthermore, stimulation with spores and germ tubes of R. arrhizus induced the upregulation of the glycoprotein CD154 (an indicator of immune cell activation) in CD4+ T cells of healthy donors [33].

3. Finally, it may be worth at least mentioning B cells and antibodies and any putative role(s) for these during fungal infection in general as this arm of the immune response is not discussed, and this could be viewed as an oversight by the authors by a broader readership.

Yes, We agree with the reviewer’s opinion. We have included a new chapter # 11 entitled ‘B cell lymphocytes and humoral response against Mucorales’ highlighting the role of B cells in antifungal immunity. Page 10, lines 303-316:

In contrast to T cells, B cells undergo clonal expansion and secrete specialized proteins known as antibodies (immunoglobulins) upon activation [61]. These antibodies bind to antigens expressed on the surface of pathogenic microorganisms and exert anti-microbial activity via neutralization, opsonization, complement activation, and antibody-dependent cellular cytotoxicity [61]. The importance of immunoglobulins in antifungal immunity has been controversially discussed. While some pioneering studies in the field claimed that antibodies do not play a major role in antifungal clearance, more recent studies showed that immunoglobulins confer protection against Cr. neoformans, C. albicans, Pneumocystis spp., and H. capsulatum [65-68]. Nevertheless, studies on the B cell-mediated antifungal response are still scarce. The spore protein CotH3 binds to the GRP78 receptor in endothelial cells and was reported to contribute in humoral response against Mucorales [69]: Polyclonal antibodies raised against peptides of CotH3 protected mice under neutropenia and diabetic ketoacidosis from R. delemar, Rhizomucor, Apophysomyces, Lichtheimia, Cunninghamella, and M. circinelloides [69].

Reviewer 2 Report

The review article by Montano and Voigt is a very important contribution to the field of anti-fungal immunity. The impact of fungal infections and commensalism has been underestimated over the last few years despite their impact for human health. The topic is therefore of high relevance. The role of Mucorales in shaping the immune system and immunity is nicely dissected and fills a gap in the current literature.

Minor comments:

  • Many fungi, especially candida, have been associated with pathogenicity as well as commensalism. The authors might want to comment a bit more on potential dichotomous roles of Mucorales. Could it have positive effects on shaping the immune system, i.e. training the immune system to protect from alternative infectious agents? 
  • Can Mucorales antigens also play a role in allergies?
  • Recently, the role of the tissue microenvironment for modulating immune responses has been stressed. Sodium chloride, which increases upon enhanced dietary intake, can, for example, increase Th17 cell differentiation therefore potentially Candida albicans clearance in the skin. It has been shown (Matthias et al JCI 2020) that NaCl can induce anti-inflammatory functions in bacteria specific Th17 cells but not candida specific Th17 cells. Likewise, NaCl increases Th2 cell differentiation and might therefore promote allergies. The authors should add a chapter commenting on these environmental factors that could shape anti-fungal host responses. 
  • Likewise, the cytokine microenvironment can modulate anti-fungal immune responses. Absence of IL-1b in the environment can induce anti-inflammatory candida specific T cells, whereas its presence induces pathogenic Th17 cells from the same naive T cell precursors (Nature, 2012, Noster et al, JACI 2015). The authors may discuss similar scenarios (cytokine switch factors for anti-fungal immune responses) in the context of mucorales. 

Author Response

1. Many fungi, especially Candida, have been associated with pathogenicity as well as commensalism. The authors might want to comment a bit more on potential dichotomous roles of Mucorales. Could it have positive effects on shaping the immune system, i.e. training the immune system to protect from alternative infectious agents?

We thank the reviewer for raising this important point. A total of 26 species were described to be opportunistic human pathogens among the order Mucorales. All of them are environmental and are ubiquitous. Mucor circinelloides was reported to alter the gastrointestinal microbiota in a murine model (Mueller et al. 2019, The Journal of Microbiology 57(6): DOI: 10.1007/s12275-019-8682-x.). As with other pathogenic fungi, mucoralean fungi cause fungal infections in man, mostly under immunocompromising conditions. Under immunocompetent conditions, however, these fungi shape the immune system and its plasticity by training it towards protection to other infectious agents. As an example, a recent study has indicated a significant correlation between A. fumigatus, one of the most common pathogenic fungi, and Mucorales specific-T cell numbers in healthy donors, which suggest a possible cross-reactivity of T cells against multiple fungi. This topic is addressed in the 10th chapter, Pages 9-10, lines 285-295. Reference 39 https://doi.org/10.1016/j.ijmm.2018.09.002

2. Can Mucorales antigens also play a role in allergies?

Yes, for example, in a group of allergic patients 40% of them showed Mucor sensitivity by specific IgE RAST (Aas, 1980. Allergy) DOI: 10.1111/j.1398-9995.1980.tb01791.x Moreover, Asthmatic patients showed low Mucor sensitization on skin testing (Beaumont, 1985. Allergy) DOI: 10.1111/j.1398-9995.1985.tb00214.x. Mucorales have also been involved in fatal anaphylaxis from the ingestion of mold-contaminated food (Bennett, 2011. Case Reports) DOI: 10.1097/00000433-200109000-00019.

3. Recently, the role of the tissue microenvironment for modulating immune responses has been stressed. Sodium chloride, which increases upon enhanced dietary intake, can, for example, increase Th17 cell differentiation therefore potentially Candida albicans clearance in the skin. It has been shown (Matthias et al. JCI 2020) that NaCl can induce anti-inflammatory functions in bacteria specific Th17 cells but not candida specific Th17 cells. Likewise, NaCl increases Th2 cell differentiation and might therefore promote allergies. The authors should add a chapter commenting on these environmental factors that could shape anti-fungal host responses.

We are thankful to this reviewer since this information clearly improved the current manuscript. We have included a new chapter # 12 entitled ‘The tissue microenvironment and its role in the inflammatory response’ which highlights the role of microenvironmental factors in shaping anti-fungal immunity. Page 10-11, lines 318-323:

Microenvironmental factors other than cytokines (e.g. oxygen, lipids, glucose, microbial products, or salts) are involved in the differentiation of immune cells, which might impact their capacity to control fungal infections [70-74]. For example, sodium chloride (NaCl) is usually found in peripheral tissues and promotes differentiation of human and murine Th17 cells with anti-inflammatory properties [70, 75]. The NaCl effect is reversed by the cytokine IL-1β and showed to be infection-dependent manner by promoting anti-inflammatory functions in Staphylococcus aureus-specific Th17 cells but not in Candida-specific Th17 cells [75]. In the context of Mucorales stress response to the host-microenvironment, R. arrhizus, and other Zygomycetes (former class Phycomycetes) exhibited stress tolerance after exposure to high NaCl concentrations [76, 77]. Therefore, we hypothesize that a series of factors, such as NaCl-mediated anti-inflammatory responses, high osmotic tolerance of Mucorales, and the disbalance of NaCl metabolism in people with risk to develop mucormycosis (e.g. diabetes) may create an optimal microenvironment for cutaneous mucormycosis [78-80]. Nevertheless, more studies are needed to understand the contribution of sodium chloride in shaping immune responses in the pathogenesis of the Mucorales.

4. Likewise, the cytokine microenvironment can modulate anti-fungal immune responses. Absence of IL-1b in the environment can induce anti-inflammatory candida specific T cells, whereas its presence induces pathogenic Th17 cells from the same naive T cell precursors (Nature, 2012, Noster et al, JACI 2015). The authors may discuss similar scenarios (cytokine switch factors for anti-fungal immune responses) in the context of Mucorales.

Yes, we agree with the reviewer’s opinion. We have addressed the influence of the cytokine-mediated microenvironment on the antifungal immune response in the 10th chapter, pages 9 and 10, lines 245-301. However, in the context of Mucorales, the modulation of the anti-fungal response by the cytokine microenvironment is very scarce.

Again and on behalf of the 1st author, we appreciate the valuable comment of the reviewer 2 and look very much forward to the successful publication of this mini-review.